# Mentality and behavior in COVID-19 emergency status in Japan: Influence of personality, morality and ideology

Kun Qian[1,2]*, Tetsukazu Yahara[2,3]

**1** Institute of Decision Science for a Sustainable Society, Kyushu University, Fukuoka, Japan, **2** Research Department, Kyushu Open University, Fukuoka, Japan, **3** Department of Science, Kyushu University, Fukuoka, Japan

* qk@kyudai.jp

**Data Availability Statement:** Data are available upon request due to data sharing restrictions imposed by The Ethics Committee for Psychological Studies at the Institute of Decision Science for a Sustainable Society, Kyushu University, involving participant consent and data

## Abstract

The COVID-19 pandemic began in December 2019 and severely influenced society. In response, the Japanese government declared a state of emergency on 7th April in seven prefectures. The study conducted an immediate survey on 8th April to record the response of the general public to the first emergency status due to epidemics. The study hypothesized that personality traits, moral foundation, and political ideology can influence people's mentality, cognition, and behavior toward COVID-19. Based on a nationwide dataset of 1856 respondents (male = 56.3%, $M_{age}$ = 46.7, emergency regions = 49.9%), the study found that personality, morality, and ideology altered mental health status and motivated behaviors toward COVID-19. Neuroticism and avoiding harm involved cognition and behavior through various means. The study also found significant differences among demographic groups. Results are informative and contributive to the governance and management of, and aid for, individual responses to the COVID-19 pandemic.

## Introduction

The outbreak of the 2019 coronavirus disease (COVID-19) rapidly progressed into a world-wide pandemic within a span of several months. In Japan, the first case of respiratory infection by COVID-19 was reported on 16th January. With the comparatively slow progress in Japan, the government declared a state of emergency on 7th April in seven prefectures (first level of administrative division) when the nationwide number of infection reached 4,000. Special psychobehavioral characteristics and social issues of the Japanese surfaced during the COVID-19 pandemic. The majority practice good hygiene habits and preventive measures against influenza or common colds, such as regular wearing of masks and social distancing [1, 2, 3]. However, increased social activities in April will create a major challenge in preventing the spread of the COVID-19 epidemic in Japan.

Studies on mental health, attitude, and preventive behavior toward COVID-19 were conducted in several countries. China's Wang et al. [4] reported immediate psychological responses and associated factors using data obtained from the general population at the initial

usage. To access the data, please contact The Ethics Committee for Psychological Studies at the Institute of Decision Science for a Sustainable Society, Kyushu University via Ms. Bonkohara, ketsudan[at]jimu.kyushu-u.ac.jp.

**Funding:** This study is supported by JSPS KAKENHI #17H06342, #20K03479 to KQ, and by Kyushu Open University to TY. The funders had no role in study design, data collection and analysis, decision to publish, or preparation of the manuscript. Funder website: https://www.jsps.go.jp/english/index.html.

**Competing interests:** The authors have declared that no competing interests exist.

stage of the epidemic followed by a longitudinal study that showed a significant reduction in psychological impact after one month [5]. Rajkumar [6] reviewed studies on the mental health of general public and medical staff during the COVID-19 pandemic. Shigemaru et al. [7] identified public responses and mental health modulated by COVID-19 in Japan from a predictive aspect. However, evidence remains necessary to demonstrate the effectiveness of predictions and reveal the mental status of the Japanese, especially after the emergency status.

Moreover, investigating the underlying psychological mechanisms that influence mental health and determine perception, attitude, and behavior toward COVID-19 is important. Neuroticism is a psychological factor of profound public health significance and a predictor of various mental and physical disorders [8]. Neuroticism is a component of the five-factor model of personality domains [9]. This five-factor model, also known as Big Five personality traits or Big Five model, was extensively verified by large and various demographic groups: Soto et al. demonstrated this model by a major cross-sectional survey with more than 1.2 million samples [10]. In Japan, the Big Five model has also been verified by large-sized samples [11]. Besides these quantitative cross-sectional studies, the rationality of the Big five model has been verified considering the aspects of evolution and development [12, 13]. Among the personality models, the Big Five model was most widely accepted in academics, business management, and by the general public [14, 15, 16]. The relationship between the Big Five personality traits and public health policy was widely explored [17]. Recent studies using the Big Five model and other indicators explored how personality predicted health behaviors, such as social distancing and individual hygiene during the COVID-19 pandemic, using data obtained from the United States [18] and Qatar [19]. However, they overlooked the influence of personality traits on mental health status.

Morality is another determinant of behavior toward COVID-19. Malm et al. [20] cited healthcare workers on duty during pandemic scenarios. In the COVID-19 context, moral injury was widely reported for healthcare workers [21, 22] and in actual medical scenes [23]. An experimental study on the general public demonstrated the influence of morality on preventive behaviors using deontological, virtuous, and utilitarian moral messages [24]. However, evidence remains debatable [25]. Thus, exploring how basic moral foundations alter mentality and behavior during pandemics is relevant.

Ideology is an important factor that affects thoughts and actions. Previous studies in the United States showed that political ideology influenced concerns and behaviors toward COVID-19 [26] and trust in science agencies [27]. In addition, conservatives were less concerned about COVID-19 than liberals were [28]. Thus, the present study proposes that ideologies lead to various attitudes and behaviors toward key measures against COVID-19, including the declaration of emergency status, which is a part of political governance. This hypothesis requires support in Japan, where the conservatives held the reins of government for a long time.

The present study investigates the influence of personality, morality, and ideology on mental status and the attitude, undertaking, and behavior of the Japanese toward COVID-19. First, the present study aimed to clarify the impact of personality, morality, and ideology on the citizens' mentality, opinion, and behavior during the COVID-19 crisis, wherein we predicted that certain factors related to personality, morality, and political ideology affect the mental health status, opinion, and preventive behavior of COVID-19 patients. Second, it explored the differences between demographic groups for mentality, opinion, and behaviors, wherein we considered that certain sociodemographic characteristics exert significant differences on the mental health status, opinion, and preventive behavior of COVID-19 patients. Nevertheless, as this is the first emergency situation related to public health in Japan, and the present study is the first study to explore the impact of personality, morality, and ideology on the mentality and behavior of Japanese people, no specific hypotheses have been set for these two research objectives.

We presume that the outcomes of this exploratory study will be informative and contributive to the governance and management, as well as individuals' mental healthcare and preventive behavior, not only during the ongoing COVID-19 crisis, but also during any emergency related to public health event in future.

## Methods

### Ethical information

Expedited ethical approval for the study was obtained from the Ethics Committee for Psychological Studies at the Institute of Decision Science for a Sustainable Society, Kyushu University (No. 2020/1-7). All methods employed were conducted in accordance with the relevant guidelines of the ethics committee and the code of ethics and conduct of the Japanese Psychological Association and Declaration of Helsinki. The survey was conducted anonymously. The study protocol and data using policy were disclosed at the recruitment page as well as the beginning of the questionnaire. The questionnaire survey commenced only after the participant accepted the data using policy and agreed to participate.

### Participants and procedure

We conducted a cross-sectional survey online through Yahoo! Crowdsourcing service (operated by Yahoo Japan Corporation; hereafter referred to as Yahoo). The respondents were registered Yahoo users and were randomly selected from all prefectures in Japan. All respondents joined the survey online using Internet browsers installed in their devices, such as computers, tablets, and smartphones. Each respondent who completed the survey was paid 7 T-points, which equals seven Japanese yen via Yahoo. Voluntary respondents were also encouraged to join the survey without payment. The survey was programmed and conducted by jsPsych [29]. All the data obtained were automatically uploaded on our server at the end of the survey.

The survey was submitted to Yahoo on 7th April, the day that the Japanese government first declared a state of emergency against the 2020 COVID-19 pandemic. On the same day, seven prefectural divisions in Japan, namely, Tokyo, Osaka, Chiba, Kanagawa, Saitama, Hyogo, and Fukuoka, entered a state of emergency. After screening by Yahoo, the survey started online at 8:00 am on 8th April and automatically ended at 19:55 on 9th April after reaching the targeted number of samples (2000 respondents with payment). The target sample size was determined via the following process. First, we conducted *a priori* power analyses by using G $^*$ Power [30]. We planned to perform t-test, one-way analysis of variance (ANOVA), and linear multiple regression. The required sample sizes were estimated as 788 for t-test ($d = 0.2$, $\alpha = 0.05$, $1 - \beta = 0.8$), 969 for one-way ANOVA with three groups ($f = 0.1$, $\alpha = 0.05$, $1 - \beta = 0.8$), and 904 for linear multiple regression with 13 predictors ($f^2 = 0.02$, $\alpha = 0.05$, $1 - \beta = 0.8$). As stated later in Survey development and Data analysis, the planned 13 predictors include 5 factors for personality, 5 for morality, and 3 for ideology. We used smaller effect sizes due to the potential data noise of online survey. Second, considering the potential abnormal and/or satisficing data [31], we doubled the biggest required sample size, 969, and eventually determined the target sample size as 2000.

### Survey development

The study aims to explore whether or not personality traits, moral foundation, and ideology can predict the resulting mentality and behavior during a state of emergency. Thus, a structured questionnaire was developed with five sections, namely, demographic data (Q1), scales of personality traits (Q2), self-reported concerns and behavior about COVID-19 (Q3), scales

of mental health status (Q4), and scales of moral foundation and ideology (Q5). The questionnaire consists of a total of 170 items. An additional 19 questions regarding knowledge of COVID-19 were included in the survey; however, they will be analyzed in a separate study. The details of the aforementioned five sections are as follows.

Demographic data included age (Q1-1), gender (Q1-2), marital status (Q1-3), parental status (Q1-4), household size (Q1-5), employment status (Q1-6), education (Q1-7), birthplace (Q1-8), and place of residence (Q1-9) and postcode (Q1-10). Postcode information was collected to verify data reliability.

The Japanese version of the Big-Five scale (BFS; [32]), one of the most reliable and popular scales for personality traits in Japan [33], was used for the scales of personality traits. BFS included an adjective checklist with 60 items related to personalities (Q2-1 to Q2-60). The participants were requested to select their answer for each item using a 7-point Likert-type scale ranging from "7 = *totally describes me*" to "1 = *does not describe me at all*". The Data analysis section will report the method used to validate the scales.

The section for self-reported concerns and behavior about COVID-19 consisted of 44 items. Nineteen items were replicated from a previous research in China [4], namely, symptoms (Q3-1) and treatment experience (Q3-2) in the past 14 days, self-rated health status (Q3-3), status of insurance (Q3-4), contact with COVID-19 cases (Q3-5 = close contact; Q3-6 = indirect contact; Q3-7 = contact with suspected infections or infected materials), route of transmission (Q3-8 = droplets; Q3-9 = contaminated objects; Q3-10 = airborne), main source of (Q3-14) and satisfaction with (Q3-15) disclosed health information, confidence in doctors (Q3-16), likelihood of infection (Q3-17) and survival (Q3-18), concerns about infections among family members (Q3-19) especially young children (Q3-20), and preventive behaviors, such as avoiding sharing of tableware (Q3-25) and handwashing with soap (Q3-26). Nine items were modified from those employed by Wang et al. [4] to fit the current situation of Japan, namely, understanding why the population of infections increased (Q3-11), death (Q3-12), cure (Q3-13), preventive behaviors, such as covering mouth with masks or arms when coughing or sneezing (Q3-24), wearing masks when speaking to people regardless of absence of symptoms (Q3-30), and washing hands after touching objects possibly touched by infected persons (Q3-33), perceived exaggerated response of society to COVID-19 (Q3-34), average number of hours away from home or facilities for medical treatment or health observation (Q3-35), and health information provided (Q3-36). The following original questions were also formulated considering the situation and needs in Japan: evaluating the response and measures of the Japanese government (Q3-21) and local municipalities (Q3-22) and preventive actions of the Japanese people (Q3-23), reporting preventive behaviors, such as disinfecting with ethanol (Q3-27), wiping off water after handwashing (Q3-28), avoiding rubbing nose and eyes (Q3-29), washing hands after touching objects touched by unspecified people (Q3-32), evaluating the sufficiency of supplies of preventive goods, such as masks and ethanol (Q3-36), daily necessities, such as toilet paper (Q3-37) and food (Q3-38), reporting the extent of the influence of the pandemic on daily life (Q3-39) and work (Q3-40), and reporting on the sufficiency of PCR tests (Q3-42), medical staff (Q3-43), and medical facilities/equipment (Q3-44). An attention check question (Q3-31) was also added in this section.

The study employed the Japanese version of the Depression, Anxiety, and Stress Scale (DASS) to collect data on mental health status. Previous research has used the scale to measure the mental status of the general public under pandemic conditions [34, 4]. The DASS includes 21 questions (i.e., Q4-1 to Q4-21). The participants were required to answer the questions using a 4-point Likert-type scale ranging from "0 = *does not describe me at all*" to "3 = *totally describes me*." The Data analysis section will also describe the method used to validate the scale.

The last section tested the scales of moral foundation and ideology. The Japanese version of the Moral Foundation Questionnaire (MFQ, Q5-1 to Q5-30) and the scale of ideology (Q5-31 to Q5-35) validated by Murayama and Miura [35] were used. The relevance items for MFQ (Q5-1 to Q5-15) required responses using a 5-point Likert-type scale ranging from "0 = It does not matter at all (It has nothing to do with the judgment)" to "4 = It is strongly taken into account (It is crucial for the judgment)" [36]. The judgment items for MFQ (Q5-16 to Q5-30) and the first four items in the scale for ideology required responses using a 5-point Likert-type scale ranging from "0 = *Completely disagree*" to "4 = *Completely agree*." The last item for the scale of ideology was a self-rated political ideology ranging from 0 (liberalism) to 10 (conservatism) with an additional option of "I don't know".

## Data analysis

Before statistical analysis, the researchers screened non-normal respondents using the following steps: evaluating answers to the attention check question (Q3-31), verifying that the postcode (Q1-10) provided exists and matches with the residence (Q1-9), and ensuring that all responses in the same questionnaire page (Q1 to Q5 were divided into 8, 3, 11, 2, and 5 pages, respectively) were given different values ($SD \neq 0$, the SD check target was Q2, Q4, and Q5). All data from non-normal respondents were excluded from statistical analysis. We referred to several previous studies for the data exclusion method [37, 38, 39]. To ensure the data quality, we excluded these abnormal data based on quite serious criteria.

After eliminating invalid data, the study validated the three cited psychological scales, namely, BFS (Q2), DASS (Q4), and MFQ (Q5), by conducting reliability analysis based on Cronbach's alpha [40] and confirmatory factor analysis. After confirming reliability, the mean response values for each subscale were calculated (BFS = extraversion, neuroticism, openness, conscientiousness, and agreeableness; DASS = stress, anxiety, and depression; and MFQ = harm (avoiding harm), fairness, ingroup loyalty, authority (respect for authority), and purity. Thus, response data of the 111 questions were grouped into 13 sets of values corresponding to the representative values for each subscale. In addition, the mean values of Q6-31 and Q6-32 were calculated as indicators of preference for equality, whereas those for Q6-33 and Q6-34 denote an attitude that is anti-change.

The quantitative data collected in the Q3 section were also summarized by calculating the mean values for the following indicators: epidemic consciousness (Q3-11 to Q3-13), evaluation of others (Q3-21 to Q3-23), preventive e behavior (Q3-24 to Q3-30, Q3-32, and Q3-33), material sufficiency (Q3-36 to Q3-38), and medical sufficiency (Q3-42 to Q3-44). Other quantitative data (Q3-3, Q3-15 to Q3-20, Q3-34, Q3-39, and Q3-40) as well as the data of most categorical variables (Q3-2 to Q3-10, Q3-14, Q3-35, and Q3-41) were directly used for statistical analysis without pre-processing. The demographic data in Q1 were regrouped as a preparation of the t-test and one-way ANOVAs. Table 1 provides the details of the regrouping.

After the abovementioned preparation, a series of multiple linear regression analyses, t-tests and one-way ANOVAs were performed to examine the hypotheses. To explore the first research objective, the subscales personality traits (Q2) and moral foundation and ideology (Q5) were introduced as predictor variables. Concerns and behavior (Q3) and mental status (Q4) were assigned as dependent variables. For the second research objective, a series of simple regression analysis, t-test, and one-way ANOVA was carried out using the items of Q1 as independent variables, and those of Q3 and Q4 as dependent variables.

Detection and elimination of non-normal respondents and calculation of the mean values and *SD* were performed using Microsoft Excel for Mac (Version 16.35). Reliability analysis of the scales and simple/multiple linear regression analyses were conducted using IBM SPSS

**Table 1. Sociodemographic characteristics and numbers of samples (n = 1856).**

| Demographics | Options | N | (%) |
|---|---|---|---|
| *Gender* | Male | 1044 | 56.3 |
| | Female | 809 | 43.6 |
| | Other | 3 | 0.2 |
| *Marital status* | Unmarried | 719 | 38.7 |
| | Married | 1013 | 54.6 |
| | Divorced / widowed | 124 | 6.7 |
| *Parental status* | No children | 997 | 53.7 |
| | Have children aged 16 or under | 416 | 22.4 |
| | All children aged 17 or over | 443 | 23.9 |
| *Household size* | 1 | 357 | 19.2 |
| | 2 | 515 | 27.7 |
| | 3 to 5 | 942 | 50.8 |
| | More than 6 | 42 | 2.3 |
| *Employment status* | Company officer / executive | 33 | 1.8 |
| | Company employee (permanent) | 608 | 32.8 |
| | Public employee (permanent) | 63 | 3.4 |
| | Teachers / researchers | 12 | 0.6 |
| | *(Summarized as Full-time employed)* | *716* | *38.6* |
| | Company employee (temporary) | 108 | 5.8 |
| | Public employee (temporary) | 4 | 0.2 |
| | Agriculture / forestry / fisheries | 11 | 0.6 |
| | Self-emplyed / freelance | 209 | 11.3 |
| | Part-time | 263 | 14.2 |
| | Work at home | 14 | 0.8 |
| | *(Summarized as Part-time / self employed)* | *609* | *32.9* |
| | Housewife / househusband | 226 | 12.2 |
| | Student (high school or under) | 6 | 0.3 |
| | Student (college or postgraduate) | 16 | 0.9 |
| | Retired with annuity | 90 | 4.8 |
| | Unemployed | 161 | 8.7 |
| | Other | 32 | 1.7 |
| | *(Summarized as Unemployed)* | *531* | *28.6* |
| *Education* | Primary school or under | 1 | 0.1 |
| | Junior middle school | 34 | 1.8 |
| | Senior middle school (high school) | 484 | 26.1 |
| | Colleges of technology (*Kōsen* in Japanese) | 17 | 0.9 |
| | Specialised training college (*Senmon gakkō*) | 241 | 13 |
| | Junior colledge | 160 | 8.6 |
| | Other | 8 | 0.4 |
| | *(Summarized as Basicly educated)* | *945* | *50.9* |
| | Bachelor | 799 | 43 |
| | Master | 85 | 4.6 |
| | Doctorate | 27 | 1.5 |
| | *(Summarized as Highly educated)* | *911* | *49.1* |
| *Place of residence* | Prefectures in status of emergency | 926 | 49.89 |
| *(Summarized)* | *(Tokyo, Osaka, Chiba, Kanagawa, Saitama, Hyogo, and Fukuoka)* | | |
| | Other | 930 | 50.11 |

Statistics Base (Version 25). t-tests, one-way ANOVA, and post-hoc comparisons based on Tukey's method were performed using Jamovi (Version 1.2.16.0; [41, 42, 43, 44, 45]). All software was operated on Apple iMac Pro (Model A1862, macOS Catalina Version 10.15.4).

## Results

### Data collection and demographics

Data were collected from a total of 2233 respondents (i.e., 2000 rewarded and 233 voluntary). All respondents completed the questionnaires, out of which data from 377 respondents were excluded after screening for non-normal respondents (26 wrong answers to attention check questions, 87 invalid postcodes, and 264 unvaried responses to all questions in at least one of the same questionnaire page). Thus, data collected from 1856 respondents (1044 males, 809 females, and three others; mean age = 46.69 years; $SD$ = 11.29 years) were used as the final data for statistical analysis. The majority of respondents were male (56.3%), married (54.6%), without children (53.7%), with a household size of 3 to 5 (50.8%), full-time permanent employees (32.8%), and college graduates (43%). Half of the respondents were residents of the prefectures where a state of emergency has been declared (total = 49.9%; Tokyo = 11.4%, Osaka = 9%, Kanagawa = 8.8%, Saitama = 7.1%, Hyogo = 5%, Fukuoka = 4.5%, and Chiba = 4.1%). Table 1 summarizes the detailed data on sociodemographic characteristics.

### Validation of scale reliability

The results of Cronbach's alpha (α) indicated high internal consistency for each subscale under BFS (extraversion = .92, neuroticism = .94, openness = .90, conscientiousness = .90, and agreeableness = .77) and DASS (stress = .86, anxiety = .82, and depression = .90). However, the results for MFQ were lower (harm = .69, fairness = .67, ingroup loyalty = .63, authority = .61, and purity = .59) but coincident with a previous research that validated the Japanese version of MFQ [35]. For confirmatory factor analysis, Table 2 presents the results of the goodness-of-fit indices of factor models for BFS, DASS, and MFQ. All scales were well fitted to their expected factor models. These results indicate that the scales used had high internal consistency and reliability.

### Influence of personality, morality, and ideology on mentality, opinion, and preventive behavior to COVID-19

The results of multiple linear regression analyses related to our first research objective are presented in Table 3. All regression equations were significant with the 13 factors of personality, morality, and ideology as the predictors and stress, anxiety, depression, epidemic consciousness, underestimation of the pandemic, preventive behavior, material sufficiency, medical sufficiency, information sufficiency, self-related health status, likelihood of infection, likelihood

**Table 2. Goodness-of-fit indices for confirmatory factor analyses of BFS, DASS, and MFQ.**

|  | $\chi^2$ | df | p | AIC | CFI | TLI | SRMR | RMSEA |
|---|---|---|---|---|---|---|---|---|
| BFS (5 factors) | 19603 | 1700 | < .001 | 308833 | .743 | .732 | .101 | .075 |
| DASS (3 factors) | 3485 | 186 | < .001 | 69409 | .849 | .830 | .056 | .098 |
| MFQ (5 factors) | 8469 | 395 | < .001 | 160904 | .619 | .581 | .105 | .105 |

AIC = Akaike's information criterion, CFI = comparative fit index, TLI = Tucker-Lewis index, SRMR = standardized root-mean-square residual, RMSEA = root-mean-square error of approximation.

**Table 3. Results of multiple linear regression analyses ($n = 1856$).** The mean response values of the BFS and MFQ scales and ideology items are predictor variables. All independent variables were employed in the models.

| Variables | Stress B | SE B | β | | Anxiety B | SE B | β | | Depression B | SE B | β | | Epidemic consciousness B | SE B | β | | Underestimation B | SE B | β | | Preventive behavior B | SE B | β | | Material sufficiency B | SE B | β | | Medical sufficiency B | SE B | β | | Information sufficiency B | SE B | β | |
|---|---|---|---|---|---|---|---|---|---|---|---|---|---|---|---|---|---|---|---|---|---|---|---|---|---|---|---|---|---|---|---|---|---|---|---|---|---|
| *Personality* | | | | | | | | | | | | | | | | | | | | | | | | | | | | | | | | | | | | | |
| Extraversion | .057 | .018 | .089 | | .025 | .015 | .052 | | -.032 | .021 | -.045 | | .040 | .021 | .057 | | -.089 | .036 | -.079 | * | .079 | .021 | .114 | *** | -.034 | .026 | -.042 | | -.115 | .024 | -.150 | *** | -.054 | .042 | -.041 | |
| Neuroticism | .259 | .015 | .436 | *** | .141 | .012 | .312 | *** | .281 | .017 | .416 | *** | -.034 | .017 | -.052 | | -.070 | .029 | -.067 | * | .055 | .017 | .085 | ** | -.085 | .021 | -.114 | *** | -.066 | .020 | -.092 | ** | -.131 | .034 | -.106 | *** |
| Openness | .010 | .019 | .014 | | .016 | .015 | .030 | | .008 | .021 | .010 | | .122 | .022 | .162 | *** | .013 | .037 | .011 | | .054 | .022 | .072 | * | -.009 | .027 | -.010 | | .077 | .025 | .092 | ** | -.045 | .044 | -.031 | |
| Conscientiousness | .006 | .018 | .008 | | -.011 | .015 | -.021 | | -.058 | .021 | -.073 | ** | .079 | .021 | .103 | *** | -.004 | .035 | -.003 | | .103 | .021 | .135 | *** | -.011 | .026 | -.013 | | -.012 | .024 | -.014 | | .052 | .042 | .036 | |
| Agreeableness | -.115 | .026 | -.120 | *** | -.048 | .021 | -.065 | * | -.036 | .030 | -.033 | | -.033 | .031 | -.031 | | -.046 | .051 | -.027 | | .072 | .030 | .069 | * | .088 | .037 | .072 | * | .019 | .035 | .016 | | -.018 | .061 | -.009 | |
| *Morality* | | | | | | | | | | | | | | | | | | | | | | | | | | | | | | | | | | | | | |
| Harm | -.076 | .031 | -.096 | * | -.109 | .025 | -.179 | *** | -.060 | .035 | -.066 | | -.013 | .036 | -.015 | | -.150 | .060 | -.107 | * | .144 | .036 | .167 | *** | .106 | .043 | .105 | * | -.052 | .041 | -.054 | | .066 | .071 | .040 | |
| Fairness | .037 | .030 | .045 | | .029 | .024 | .045 | | .076 | .034 | .081 | * | .023 | .035 | .026 | | -.039 | .059 | -.027 | | -.026 | .035 | -.029 | | -.067 | .042 | -.064 | | -.100 | .040 | -.100 | * | -.147 | .070 | -.086 | * |
| Ingroup | .034 | .031 | .039 | | .043 | .025 | .065 | | -.008 | .035 | -.008 | | .105 | .036 | .112 | ** | .003 | .060 | .002 | | .027 | .036 | .029 | | -.049 | .044 | -.046 | | .000 | .041 | .000 | | -.001 | .071 | -.001 | |
| Authority | .061 | .032 | .069 | | .032 | .025 | .048 | | .057 | .036 | .058 | | .032 | .037 | .034 | | .080 | .061 | .052 | | -.072 | .036 | -.076 | * | .048 | .044 | .044 | | .087 | .042 | .083 | * | .037 | .072 | .020 | |
| Purity | .006 | .029 | .006 | | .005 | .024 | .007 | | -.027 | .033 | -.027 | | -.065 | .034 | -.068 | | -.079 | .056 | -.051 | | .046 | .034 | .048 | | .019 | .041 | .017 | | -.067 | .039 | -.063 | | -.021 | .067 | -.011 | |
| *Ideology* | | | | | | | | | | | | | | | | | | | | | | | | | | | | | | | | | | | | | |
| Equality | -.014 | .017 | -.021 | | -.015 | .014 | -.030 | | -.002 | .019 | -.003 | | .102 | .020 | .141 | *** | -.085 | .033 | -.072 | * | .014 | .020 | .020 | | .072 | .024 | .086 | ** | -.029 | .023 | -.036 | | .030 | .040 | .022 | |
| Antichange | .012 | .016 | .018 | | .006 | .013 | .012 | | .006 | .018 | .008 | | -.029 | .019 | -.042 | | .047 | .031 | .042 | | -.001 | .018 | -.002 | | -.035 | .022 | -.044 | | .007 | .021 | .009 | | .139 | .037 | .104 | *** |
| Political ideology | -.025 | .007 | -.086 | ** | -.013 | .006 | -.060 | ** | -.015 | .008 | -.045 | | .012 | .008 | .038 | ** | -.041 | .014 | -.081 | ** | .006 | .008 | .020 | ** | .011 | .010 | .031 | | .006 | .009 | .017 | | .063 | .016 | .106 | *** |
| $R^2$ | .112 | | | | .233 | | | | .233 | | | | .118 | | | | .046 | | | | .125 | | | | .038 | | | | .056 | | | | .112 | | | |
| F | 15.594 | | | *** | 37.483 | | | *** | 37.483 | | | *** | 16.414 | | | *** | 5.889 | | | *** | 17.540 | | | *** | 4.826 | | | *** | 7.250 | | | *** | 15.594 | | | *** |

| Variables | Self-rated Health status B | SE B | β | | Likelihood of infection B | SE B | β | | Likelihood of surviving B | SE B | β | | Evaluation to others B | SE B | β | | Confidence in doctors B | SE B | β | | Concerns on family B | SE B | β | | Concerns on children B | SE B | β | | Influence on life B | SE B | β | | Influence on work B | SE B | β | |
|---|---|---|---|---|---|---|---|---|---|---|---|---|---|---|---|---|---|---|---|---|---|---|---|---|---|---|---|---|---|---|---|---|---|---|---|---|---|
| *Personality* | | | | | | | | | | | | | | | | | | | | | | | | | | | | | | | | | | | | | |
| Extraversion | .075 | .029 | .079 | * | -.022 | .038 | -.019 | | .053 | .038 | .044 | | -.115 | .032 | -.115 | *** | -.050 | .042 | -.038 | | .090 | .033 | .086 | ** | .126 | .035 | .116 | ** | .046 | .030 | .049 | | .089 | .039 | .073 | * |
| Neuroticism | -.213 | .024 | -.240 | *** | .104 | .031 | .096 | ** | -.109 | .031 | -.098 | *** | -.130 | .026 | -.139 | *** | -.137 | .034 | -.112 | *** | .158 | .027 | .161 | *** | .076 | .028 | .075 | ** | .084 | .024 | .096 | *** | .070 | .032 | .062 | * |
| Openness | -.038 | .030 | -.037 | | .004 | .039 | .003 | | -.001 | .040 | -.001 | | -.006 | .033 | -.005 | | -.051 | .043 | -.036 | | -.093 | .034 | -.081 | ** | -.005 | .036 | -.004 | | .047 | .031 | .046 | | .126 | .040 | .095 | ** |
| Conscientiousness | .043 | .029 | .041 | | -.100 | .037 | -.078 | ** | -.087 | .038 | -.067 | * | -.083 | .031 | -.075 | ** | -.034 | .041 | -.024 | | -.002 | .033 | -.002 | | -.048 | .034 | -.040 | | .006 | .030 | .006 | | -.082 | .039 | -.061 | * |
| Agreeableness | .115 | .042 | .080 | ** | .041 | .054 | .023 | | .240 | .055 | .132 | *** | .153 | .045 | .101 | ** | .188 | .060 | .095 | ** | .073 | .047 | .046 | | .060 | .050 | .037 | | .070 | .043 | .049 | | .109 | .056 | .059 | |
| *Morality* | | | | | | | | | | | | | | | | | | | | | | | | | | | | | | | | | | | | | |
| Harm | .043 | .049 | .036 | | .161 | .063 | .110 | * | .088 | .065 | .059 | | .092 | .053 | .074 | | .056 | .070 | .035 | | .294 | .055 | .224 | *** | .172 | .058 | .127 | ** | .186 | .050 | .159 | *** | .064 | .066 | .042 | |
| Fairness | -.088 | .048 | -.072 | | -.014 | .062 | -.009 | | -.056 | .063 | -.036 | | -.159 | .052 | -.123 | ** | .026 | .069 | .015 | | -.012 | .054 | -.009 | | -.026 | .057 | -.018 | | .053 | .049 | .044 | | .029 | .064 | .019 | |
| Ingroup | .053 | .049 | .041 | | .008 | .063 | .005 | | .015 | .065 | .009 | | .110 | .053 | .081 | * | .040 | .070 | .023 | | .097 | .056 | .068 | | .206 | .059 | .141 | *** | -.015 | .050 | -.012 | | .090 | .066 | .055 | |
| Authority | .030 | .050 | .023 | | -.071 | .064 | -.044 | | -.078 | .066 | -.048 | | -.027 | .054 | -.019 | | -.082 | .071 | -.046 | | -.023 | .056 | -.016 | | -.101 | .059 | -.068 | | -.021 | .051 | -.016 | | -.031 | .067 | -.019 | |
| Purity | .027 | .046 | .020 | | .055 | .059 | .034 | | .044 | .061 | .026 | | -.054 | .050 | -.039 | | -.030 | .066 | -.016 | | -.120 | .052 | -.083 | * | -.083 | .055 | -.055 | | .000 | .047 | .000 | | .050 | .062 | .030 | |
| *Ideology* | | | | | | | | | | | | | | | | | | | | | | | | | | | | | | | | | | | | | |
| Equality | -.012 | .027 | -.012 | | .060 | .035 | .049 | | -.030 | .036 | -.024 | | .044 | .030 | .042 | | .161 | .039 | .118 | *** | .044 | .031 | .040 | | -.021 | .032 | -.019 | | -.014 | .028 | -.015 | | .023 | .037 | .018 | |
| Antichange | .020 | .025 | .021 | | -.020 | .033 | -.017 | | .003 | .033 | .002 | | .074 | .027 | .073 | ** | .033 | .036 | .025 | | .007 | .029 | .007 | | .065 | .030 | .060 | * | -.013 | .026 | -.014 | | -.050 | .034 | -.041 | |
| Political ideology | -.023 | .011 | -.055 | * | -.003 | .015 | -.006 | | .006 | .015 | .012 | | .055 | .012 | .121 | *** | .033 | .016 | .056 | * | -.019 | .013 | -.040 | | .002 | .013 | .004 | | .007 | .012 | .017 | | .005 | .015 | .010 | |
| $R^2$ | .112 | | | | .039 | | | | .039 | | | | .070 | | | | .046 | | | | .086 | | | | .044 | | | | .056 | | | | .046 | | | |
| F | 15.594 | | | *** | 5.057 | | | *** | 4.933 | | | *** | 9.229 | | | *** | 5.889 | | | *** | 11.560 | | | *** | 5.713 | | | *** | 7.243 | | | *** | 5.936 | | | *** |

\*\*\* $p < .001$
\*\* $p < .01$
\* $p < .05$

of survival, evaluation to others, confidence in doctors, concerns regarding family and children, influence on life on work as the dependent variables.

Notably, the mentality, opinion, and behavior related to COVID-19 were fairly predicted by the personalities using several aspects. Extraversion was revealed as a significant predictor of underestimation, medical sufficiency, evaluation to others, and a positive predictor of preventive behavior, self-rated health status, concerns regarding family and children, and influence on work. Neuroticism is a significant positive predictor of stress, anxiety, depression, preventive behavior, likelihood of infection, concerns regarding family and children, influence on life and work, and a negative predictor of underestimation, material sufficiency, medical sufficiency, information sufficiency, self-rated health status, likelihood of surviving, evaluation to others, and confidence in doctors. Openness is a significant positive predictor of stress, epidemic consciousness, preventive behavior, medical sufficiency, influence on work, and negative predictor of concerns regarding family. Conscientiousness was revealed as a significant negative predictor of depression, likelihood of infection, evaluation to others, and influence on work, and as a positive predictor of epidemic consciousness and preventive behavior. Agreeableness is a significant negative predictor of stress and anxiety, and a positive predictor of preventive behavior, material sufficiency, self-rated health status, likelihood of surviving, evaluation to others, and confidence in doctors.

For factors of morality, the moral foundation of "harm," which denotes avoiding harming others and providing care and protection, negatively influenced stress, anxiety, underestimation of the pandemic, and positively influenced the preventive behavior, material sufficiency, likelihood of infection, concerns regarding family and children, and influence on life. Fairness positively contributed to depression and negatively to medical sufficiency, information sufficiency, and evaluation to others. Ingroup loyalty was revealed as a positive predictor of epidemic consciousness, evaluation to others, and concerns regarding children. Respect for authority revealed a significant negative regression with preventive behavior and positive regression with medical sufficiency. Purity indicated a negative regression with concerns about family.

For factors of ideology, preference for equality was a significant positive predictor of epidemic consciousness, material sufficiency, confidence in doctors, and a negative predictor of underestimation. Resistance to change was defined as a significant positive predictor of information sufficiency, evaluation to others and concerns regarding children. It was also revealed that conservative ideology revealed a significant negative regression with stress, anxiety, underestimation, and self-rated health status, and a positive regression with information sufficiency, evaluation to others, and confidence in doctors.

## Association of mentality, opinion, and preventive behavior to COVID-19 with demographic characteristics

For the second research objective, we investigated the association of demographic characteristics with the concerns and preventive behavior toward COVID-19. Age was the only quantitative data in Q1; thus, simple regression analyses were run with age as a predictor. Age had a significant negative effect on stress ($F$ (1, 1854) = 36.978, $p < .001$; $R^2 = .019$, $\beta = -.140$), anxiety ($F$ (1, 1854) = 15.463, $p < .001$; $R^2 = .008$, $\beta = -.091$), depression ($F$ (1, 1854) = 38.066, $p < .001$; $R^2 = .020$, $\beta = -.142$), preventive behavior ($F$ (1, 1854) = 4.511, $p = .034$; $R^2 = .002$, $\beta = -.049$), medical sufficiency ($F$ (1, 1854) = 8.821, $p = .003$; $R^2 = .005$, $\beta = -.069$), likelihood of infection ($F$ (1, 1854) = 15.610, $p < .001$; $R^2 = .008$, $\beta = -.091$) and survival ($F$ (1, 1854) = 7.388, $p = .007$; $R^2 = .004$, $\beta = -.063$), and concerns about family ($F$ (1, 1854) = 12.426, $p < .001$; $R^2 = .007$, $\beta = -.082$) and children ($F$ (1, 1854) = 9.655, $p = .002$; $R^2 = .005$, $\beta = -.072$).

Age had a significant positive effect on epidemic consciousness ($F$ (1, 1854) = 39.152, $p <$ .001; $R^2$ = .021, $\beta$ = .144), material sufficiency ($F$ (1, 1854) = 11.181, $p$ = .001; $R^2$ = .006, $\beta$ = .077), and confidence in doctors ($F$ (1, 1854) = 14.249, $p <$ .001; $R^2$ = .008, $\beta$ = .087). The results of the t-test denoted that demographic characteristics with two states also influenced the concern and preventive behavior to COVID-19 (Table 4). For gender, the male respondents obtained higher scores in epidemic consciousness, evaluations of others, medical sufficiency, confidence in doctors, and underestimation than female respondents. Conversely, the female participants obtained higher scores in preventive behavior, health status, likelihood of infection, and concerns about family and children than male. In terms of marital status, unmarried, divorced, or

**Table 4. Significant differences ($ps <$ .05) revealed by t-test ($n$ = 1856) with demographic characteristics as dependent variables.**

| Variables | Group A | | Group B | | $p$ | Cohen's $d$ |
|---|---|---|---|---|---|---|
| | **M** | **SE** | **M** | **SE** | | |
| *Gender (Q1-2)* | Male ($n$ = 1044) | | Female ($n$ = 809) | | | |
| Epidemic consciousness | 1.826 | .021 | 1.738 | .022 | .004 | .136 |
| Evaluation to others | 1.272 | .029 | 1.166 | .032 | .013 | .116 |
| Preventive behavior | 2.578 | .020 | 2.940 | .019 | < .001 | -.598 |
| Medical sufficiency | 0.589 | .023 | 0.487 | .024 | .002 | .144 |
| Health status | 2.653 | .028 | 2.743 | .030 | .029 | -.102 |
| Confidence in doctors | 2.318 | .036 | 2.069 | .042 | < .001 | .211 |
| Likelihood of infection | 2.518 | .034 | 2.630 | .036 | .024 | -.106 |
| Concerns on family | 3.088 | .031 | 3.336 | .032 | < .001 | -.262 |
| Concerns on children | 2.697 | .031 | 2.909 | .035 | < .001 | -.214 |
| Underestimation | 0.687 | .034 | 0.451 | .031 | < .001 | .233 |
| *Marital status (Q1-3)* | Unmarried or divorced ($n$ = 843) | | Married ($n$ = 1013) | | | |
| Stress | 0.730 | .021 | 0.663 | .018 | .015 | .114 |
| Anxiety | 0.359 | .016 | 0.313 | .014 | .027 | .103 |
| Depression | 0.887 | .025 | 0.662 | .019 | < .001 | .343 |
| Epidemic consciousness | 1.721 | .023 | 1.845 | .020 | < .001 | -.191 |
| Preventive behavior | 2.704 | .022 | 2.765 | .020 | .039 | -.097 |
| Material sufficiency | 1.890 | .026 | 1.972 | .023 | .016 | -.112 |
| Health status | 2.597 | .032 | 2.771 | .026 | < .001 | -.199 |
| Concerns on family | 3.107 | .035 | 3.268 | .028 | < .001 | -.168 |
| Concerns on children | 2.563 | .032 | 2.977 | .032 | < .001 | -.427 |
| *Education (Q1-7)* | Basicly educated ($n$ = 945) | | Highly educated ($n$ = 911) | | | |
| Epidemic consciousness | 1.722 | .021 | 1.858 | .022 | < .001 | -.210 |
| Evaluation to others | 1.148 | .030 | 1.304 | .030 | < .001 | -.172 |
| Material sufficiency | 1.894 | .025 | 1.977 | .023 | .015 | -.113 |
| Confidence in doctors | 2.095 | .039 | 2.324 | .038 | < .001 | -.193 |
| Likelihood of surviving | 2.145 | .037 | 2.360 | .034 | < .001 | -.199 |
| *Place of residence (Q1-9)* | Emergency regions ($n$ = 926) | | Non-emergency regions ($n$ = 930) | | | |
| Stress | 0.722 | .020 | 0.664 | .019 | .032 | .100 |
| Preventive behavior | 2.792 | .020 | 2.683 | .021 | < .001 | .174 |
| Material sufficiency | 1.850 | .025 | 2.019 | .023 | < .001 | -.231 |
| Influence on life | 3.475 | .027 | 3.356 | .030 | .003 | .138 |
| Influence on work | 3.177 | .037 | 3.071 | .037 | .041 | .095 |

widowed people felt significantly more stress, anxiety, and depression than married ones. In contrast, married people obtained higher scores in epidemic consciousness, preventive behavior, material sufficiency, health status, and concerns about family and children. For education, highly educated individuals gained high scores in epidemic consciousness, evaluation of others, material sufficiency, confidence in doctors, and likelihood of survival. For place of residence, people living in emergency regions provided high scores in stress, preventive behavior, influence on life and work and low scores in material insufficiency.

Table 5 displays significant differences based on one-way ANOVA. Multiple comparisons based on Tukey's method were conducted for three variables, namely, parental status, household size, and employment status. Significant differences were observed for parental status for the following groups: Between respondents without children and those with children aged 16 or below in terms of depression, health status, and concerns about family and children ($ps <$ .001); Between respondents without children and those with children aged 17 or above for stress, anxiety, depression, epidemic consciousness, medical sufficiency, and concerns about children ($ps <$ .05); Between respondents with children aged 16 or below and those with

**Table 5. Significant differences ($ps <$ .05) revealed by one-way ANOVA ($n$ = 1856) with demographic characteristics as dependent variables.**

| Variables | Group A | | Group B | | Group C | | $p$ | $\eta^2$p |
|---|---|---|---|---|---|---|---|---|
| | $M$ | $SE$ | $M$ | $SE$ | $M$ | $SE$ | | |
| *Parental status (Q1-4)* | No children ($n$ = 997) | | Have children 16 or under ($n$ = 416) | | All children 17 or over ($n$ = 443) | | | |
| Stress | 0.727 | .019 | 0.748 | .030 | 0.564 | .025 | < .001 | .015 |
| Anxiety | 0.352 | .014 | 0.346 | .023 | 0.283 | .020 | .021 | .004 |
| Depression | 0.871 | .022 | 0.682 | .031 | 0.601 | .027 | < .001 | .032 |
| Epidemic consciousness | 1.721 | .021 | 1.807 | .031 | 1.922 | .030 | < .001 | .016 |
| Medical sufficiency | 0.575 | .022 | 0.550 | .035 | 0.473 | .035 | .045 | .003 |
| Health status | 2.632 | .029 | 2.882 | .039 | 2.648 | .042 | < .001 | .014 |
| Likelihood of surviving | 2.245 | .034 | 2.380 | .052 | 2.142 | .053 | .006 | .006 |
| Concerns on family | 3.119 | .032 | 3.385 | .040 | 3.185 | .044 | < .001 | .012 |
| Concerns on children | 2.526 | .029 | 3.464 | .044 | 2.749 | .044 | < .001 | .142 |
| *Household size (Q1-5)* | 1 ($n$ = 357) | | 2 ($n$ = 515) | | 3 and more ($n$ = 984) | | | |
| Stress | 0.709 | .031 | 0.636 | .024 | 0.717 | .019 | .034 | .004 |
| Anxiety | 0.386 | .027 | 0.289 | .016 | 0.339 | .014 | .005 | .006 |
| Depression | 0.883 | .037 | 0.715 | .027 | 0.746 | .022 | < .001 | .008 |
| Material sufficiency | 1.866 | .041 | 2.009 | .032 | 1.921 | .023 | .013 | .005 |
| Health status | 2.591 | .051 | 2.777 | .036 | 2.684 | .028 | .008 | .005 |
| Concerns on family | 2.866 | .059 | 3.181 | .043 | 3.321 | .027 | < .001 | .032 |
| Concerns on children | 2.429 | .050 | 2.631 | .042 | 3.003 | .031 | < .001 | .057 |
| *Employment status (Q1-6)* | Full-time employed ($n$ = 716) | | Part-time/self-employed ($n$ = 609) | | Unemployed ($n$ = 531) | | | |
| Anxiety | 0.365 | .018 | 0.329 | .016 | 0.298 | .018 | .028 | .004 |
| Epidemic consciousness | 1.834 | .024 | 1.738 | .026 | 1.785 | .030 | .028 | .004 |
| Preventive behavior | 2.692 | .023 | 2.745 | .026 | 2.790 | .028 | .024 | .004 |
| Material sufficiency | 1.907 | .028 | 1.894 | .030 | 2.019 | .031 | .008 | .005 |
| Likelihood of infection | 2.645 | .039 | 2.535 | .042 | 2.499 | .048 | .037 | .004 |
| Concerns on children | 2.835 | .038 | 2.688 | .040 | 2.844 | .041 | .009 | .005 |
| Influence on work | 3.235 | .040 | 3.222 | .042 | 2.863 | .053 | < .001 | .022 |

children aged 17 or above in relation to stress, epidemic consciousness, health status, likelihood of survival, and concerns about family and children ($ps < .05$). As per household size, significant differences in anxiety, depression, material sufficiency, health status, and concerns about family and children ($ps < .05$) were noted between respondents living alone and those living with another person; in depression, concerns about family and children ($ps < .05$) between respondents living alone and those living with more than two persons; and in stress and concerns about family and children ($ps < .05$) between respondents living with another person and those living with more than two persons. In addition, significant differences in employment status were observed between full-time employees and part-time employees/self-employed in relation to epidemic consciousness and concerns about children ($ps < .05$); between full-time employees and unemployed individuals for anxiety, preventive behavior, material sufficiency, likelihood of infection, and influence on work ($ps < .05$); and between part-time employees/self-employed and unemployed individuals for material sufficiency, concerns about children, and influence on work ($ps < .05$).

## Discussion

The study explored and demonstrated the impact on citizens' mentality, opinion, and preventive behavior to COVID-19 by personality, morality, and ideology, and these varied with the demographic characteristics. Personality factors of neuroticism, openness, conscientiousness, and agreeableness, morality of avoiding harm and fairness, and political ideology significantly predicted the mental health status. All factors of personality, morality, and ideology were found as significant predictors of one or several specific opinions, concern, and behavior to COVID-19, including consciousness about and underestimation of epidemics, preventive behaviors, sufficiency of material supplies, medical measures, and disclosed information, self-rated health status and likelihood of infection and survival, evaluation of others, confidence in doctors, concerns about family and children, and perceived influence on life and work. The present study also clarified that high evaluation of the measures suggested by the government and the general public approach, sufficient material supplies, positive health status, and likelihood of survival were found as factors that relieve stress, anxiety, or depression. Concerns about family, underestimation of pandemic and perceived influence on work promoted mental burdens. In addition, opinions, concerns, and behavior to COVID-19 significantly varied with demographic characteristics, such as gender, age, marital status, education, place of residence, parental status, household size, and employment status. Results replicated those of previous studies, that is, citizens suffered from mental burden due to COVID-19 [4,6], although the number of infections was comparatively small in Japan [46]. Furthermore, results indicated that concerns, mental health status, and preventive behaviors to COVID-19 are associated with personality, morality, ideology, and demographic characteristics that have been studied independently [19] and [18] for personality; [25] and [24] for morality; [28] and [26] for ideology; [4, 5] for demography.

The results of the present study can provide informative implications to the governance and management of COVID-19. Concerning personality, the results revealed that neuroticism induced stress, anxiety, depression, dissatisfaction with material goods, medical care, disclosed information, distrust of the other people including doctors, and negative consideration of individual health status and survival. Conscientiousness and agreeableness contributed to lessening mental burden and increasing confidence about health, survival, and doctors. However, the negative coefficient of conscientiousness with evaluation of others and perceived influence on work implied a potential risk that diligent individuals might be working hard and might expect other people to work as well, even during the time of emergency. Practically, it is not

recommended for the government to identify the personality of each individual, because it is neither efficient nor realizable in the emergency situation of COVID-19. Furthermore, even if the personality of an individual could be defined, it would be difficult to change people's personalities in a short duration. Thus, the government should focus on the personality traits as well as the thoughts and behaviors due to these personalities, but not on specific individuals or groups who display accentuated relevant personality traits. General measures to restrain the personality trait of neuroticism and motivate conscientiousness and agreeableness are advisable (e.g., relieving anxiety and stress, encouraging people to take responsibility in their daily life, and demonstrating understanding and gratitude for the efforts of people during this pandemic). For morality, avoiding harm had many contributions, such as lightening mental burden, promoting estimation and behavior to prevent epidemic, and increasing concern about family and children. Avoiding harm positively predicted likelihood of infection, which suggested that people with this morality, such as medical workers, might be prepared for the worst, that is, infection. For the government, actions or measures should be considered with fairness, because the results implied that individuals regarding fairness as important morality responded with lower evaluation on sufficiency of medical cares and disclosed information, as well as measures of the government. Respect for authority showed a negative effect on preventive behavior and a positive one on medical sufficiency, although medical shortage was a consensus in Japan. This finding implied that excessive respect for authority might disturb people to act appropriately during the COVID-19 pandemic. In terms of ideology, conservative people showed less mental burden, satisfaction with disclosed information, high levels of evaluation of others and confidence in doctors, which suggested high levels of trust and confidence in the conservative government of Japan. In contrast, the government should encourage liberal people to encounter the crisis. Material sufficiency was considered helpful in reducing stress, anxiety, and depression, which indicates that ensuring material supply is crucial for public reassurance. The results also denoted that people living in emergency regions suffered from increased stress, less material supply, and more influence on life and work. Married people and parents of young children experienced heavy mental burden and concerns about family and children. By contrast, the elderly and singles reported less material and medical sufficiency, lower health status, and less preventive behavior. Thus, measures and operations specific to different demographic groups are required.

The study also provided suggestions for individuals in confronting COVID-19. The first is "remaining calm about COVID-19," because neuroticism resulted in a negative impact to mind, mentality, and behavior. Showing agreeableness, which refers to consideration and empathy for others, is also important and constructive for relieving mental burden and building self-confidence and social confidence. Maintaining a positive health status, remaining confident in survival despite being infected, and correctly recognizing COVID-19 without underestimation were also revealed as contributors that ease stress, anxiety, and depression. The significant difference in gender denoted that men behaved in a manner opposite from women. Maintaining effective communication with partners may thus be a means to share correct information and promote appropriate behaviors. Clearly, married people and people with bigger families felt less stress, anxiety, and depression compared with other groups. This result suggested that staying with a spouse or family can ease mental burden.

The study is subject to limitations. In fact, the database on the 170 questionnaire items included more information than that analyzed in the present study. Future studies are required to understand the complex interactions between personality, morality, ideology, and various demographic characteristics from different domains using several methodologies. Moreover, as indicated by an anonymous reviewer, personality, moral foundations, and ideology are not changeable in a short period. It is neither practicable nor necessary to change people. In

contrast, there is a need to develop systematic instructions on emergency status of COVID-19 and other pandemics in future to help the understanding and behavior of citizens with different personality traits, moral sense, and political ideologies. Lastly, the COVID-19 pandemic is rapidly changing. In Japan, less than ten days after data sampling, the government declared a nationwide state of emergency. Thus, a longitudinal study is recommended to reveal changes in mind, mentality, and behavior and how such aspects are mediated by personality, morality, and ideology across different phases of the pandemic.

## Acknowledgments

The authors thank Hitomi Nagao, Saki Funamoto and Ai Nagahama for helping process the statistical spreadsheet.

## Author Contributions

**Conceptualization:** Kun Qian, Tetsukazu Yahara.

**Data curation:** Kun Qian.

**Formal analysis:** Kun Qian.

**Funding acquisition:** Kun Qian, Tetsukazu Yahara.

**Investigation:** Kun Qian, Tetsukazu Yahara.

**Methodology:** Kun Qian, Tetsukazu Yahara.

**Project administration:** Kun Qian.

**Resources:** Kun Qian.

**Software:** Kun Qian.

**Supervision:** Kun Qian, Tetsukazu Yahara.

**Validation:** Kun Qian, Tetsukazu Yahara.

**Visualization:** Kun Qian.

**Writing – original draft:** Kun Qian.

**Writing – review & editing:** Kun Qian, Tetsukazu Yahara.

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
