## [Decision Letter · Decision Letter 0]

21 May 2020

PONE-D-20-12844

Mentality and behavior in COVID-19 emergency status in Japan: Influence of personality, morality and ideology

PLOS ONE

Dear Dr. Qian,

Thank you for submitting your manuscript to PLOS ONE. After careful consideration, we feel that it has merit but does not fully meet PLOS ONE’s publication criteria as it currently stands. Therefore, we invite you to submit a revised version of the manuscript that addresses the points raised during the review process.

Two reviewers addressed several major and minor concerns about your manuscript. Please respond these concerns carefully.

We would appreciate receiving your revised manuscript by Jul 05 2020 11:59PM. To enhance the reproducibility of your results, we recommend that if applicable you deposit your laboratory protocols in protocols.io, where a protocol can be assigned its own identifier (DOI) such that it can be cited independently in the future. For instructions see: http://journals.plos.org/plosone/s/submission-guidelines#loc-laboratory-protocols

We look forward to receiving your revised manuscript.

Kind regards,

Kenji Hashimoto, PhD

Academic Editor

PLOS ONE

Journal Requirements:

Reviewers' comments:

Reviewer's Responses to Questions

**Comments to the Author**

1. Is the manuscript technically sound, and do the data support the conclusions?

Reviewer #1: Yes

Reviewer #2: Partly

2. Has the statistical analysis been performed appropriately and rigorously? 

Reviewer #1: Yes

Reviewer #2: No

3. Have the authors made all data underlying the findings in their manuscript fully available?

Reviewer #1: Yes

Reviewer #2: Yes

4. Is the manuscript presented in an intelligible fashion and written in standard English?

Reviewer #1: Yes

Reviewer #2: No

5. Review Comments to the Author

Reviewer #1: This study aims to test if personality predict the mental health status, concern and consideration, preventive behavior, and knowledge acquisition of COVID-19. The methodology seems solid and data were carefully analysed, the rational for the hypotheses and implications of this study need more attention.

Major concerns

Firstly, as mentioned in the limitation of the study. this study is cross-sectional; the outcome measures might have affected the ratings of personality, moral, and ideology. It is recommended to include the follow-up data in addition to the current data set.

Secondly, I am not sure if this study “can provide informative implications to the governance and management of COVID-19”. For example, “Individuals with increased tendency for neuroticism should be given extra care…” How can the government assess residents neuroticism, and how would they approach those people high in neuroticism? Similarly, I wonder how the government can reinforce conscientiousness and agreeableness, given that personality is a trait that wouldn’t change in the short period of time.

Finally, I wonder now big five was selected as “personality” among other models. Moreover, it is not clear, in Introduction, how each personality may affect concern and consideration, preventive behavior, and knowledge acquisition of COVID-19. Please expand on the rational for the relationship between big 5 and those outcomes. On the other hand, the relationship between personality and mental health can be too obvious to replicate in this study (e.g. Table 2).

Minor concerns

I wonder how authors have developed the rational for hypothesis 2. To me the relationship would be the other way around. Related to the point above, mental health plays little role in this study if it is used as an outcome variable.

How was the target number (2000) agreed? Was it determined by the design of statistical analyses?

Reviewer #2: The author vigorously investigated the characteristics of citizens regarding COVID-19 factors. Outcomes will be beneficial for the future as intended by the author if appropriately performed. However, this study includes several issues to be discussed before verifying the results.

First, it was 7th April 2020, not March, that the Japanese government first declared a state of emergency against the 2020 COVID-190 pandemic. Was this study conducted on 8 – 9th March? The relationship between the date of survey and social events just before it is crucial to interpret the results. The author should look through the whole manuscript if there are any inconsistencies about chronological order.

The author presented three hypotheses in the last paragraph of the introductory section. However, these ideas seem too vague to be defined as each hypothesis. Most results extracted from the obtained data are possibly consistent with these ideas. For example, H1: “Personality, morality, and political ideology can predict the mental health status, concern and consideration, preventive behavior, and knowledge acquisition of COVID-19” will be supported by the data suggesting any relationship between perceptual and behavioral characteristics. Therefore, I can hardly identify this study as a confirmatory one even if the author had developed hypotheses in advance. If the author had expected any specific results before conducting this survey, describe it, like that: “participants with liberal ideology should be more concerned about COVID-19.” Otherwise, this study should be presented as an exploratory study.

Maybe relevant to the vagueness of the hypotheses, this manuscript is too long to be read straightforward. Readers hardly understand what is the point of the study. I recommend the author to examine the whole manuscript to rewrite so that readers can focus on the critical findings.

The author confirmed to have gain an informed consent from each participant in the first paragraph in the methods section. Did the participants send signed informed consent form? If so, the author should describe it as well as the way of storing the signed informed consent forms securely (actually, I guess not).

If this questionnaire survey was conducted with gathering the data anonymously, the researchers should have disclosed the study protocol and data using policy to the respondents before starting the questionnaire. In this case, respondents are deemed to have accepted the policy because they voluntarily sent their answers. It is not the same as written informed consent, but an acceptable way of study. If this study was conducted with this way, the author should describe it. Otherwise, gained data without any explanation to the respondents cannot be a material of scientific study of human subjects.

The author examined the participant’s knowledge about COVID-19 with Q5 of the questionnaire. The structure of this section contains several issues.

- What was the purpose of these quiz?

- Considering that this series was firstly developed by the author, I guess the data has not been standardized. How the author qualified the score of this section?

- Some questions of the series seem hard to determine the correct answer. Did some authorities check which answer was correct?

- The author identified the answers of “do not know” in Q5 (knowledge test) as knowledge uncertainty. In my sense, however, there were few persons to choose “do not know” when they were actually uncertain about each question, guessing considerable amount of people choose “yes” or “no” with their instinct. Therefore, I do not believe that this variable represents knowledge uncertainty of the participants. Are there any preceding studies to support the way of the author?

Overall, I cannot justify the scientific use of the results of this section, unless these issues are entirely addressed.

The author described that there were 264 unvaried responses. I was surprised to know the outcome that more than 10% of the data should be excluded. Although I have no evidence regarding the standard rate, 10% looks extraordinary (In my experience, approximately 2 – 3 %). Do the author this result can be acceptable? I am afraid the population of this survey (from Yahoo group) itself was contaminated, in other words including a considerable number of dishonest subjects. Can the author believe all adopted respondents answered the questionnaire sincerely?

Were there any solutions to exclude multiple posting of a certain person certainly? If not, the validity of this survey should be collapsed.

In summary, the major weaknesses of this manuscript are below:

- Doubtful subjects

- Unconcreted hypotheses

- Redundant presentation

6. PLOS authors have the option to publish the peer review history of their article (what does this mean?). If published, this will include your full peer review and any attached files.

Reviewer #1: No

Reviewer #2: Yes: Akihiro Shiina

---

## [Author Response · Author response to Decision Letter 0]

2 Jun 2020

Dear Reviewers,

Thank you very much for reviewing our manuscript. We have carefully revised our manuscript based on your constructive comments. We highlighted the changes made to the original version in red in the file labeled 'Revised Manuscript with Track Changes'. Our point-by-point response is shown as below. We recommend checking our detailed response in the separated file labeled 'Response to Reviewers', which should be easier to read. The line numbers are based on the 'Revised Manuscript with Track Changes'.

We hope the revised manuscript is satisfactory for the publication in PLOS ONE.

Sincerely yours,

Kun Qian, PhD

***

Response to Reviewer 1

Opening comments

This study aims to test if personality predict the mental health status, concern and consideration, preventive behavior, and knowledge acquisition of COVID-19. The methodology seems solid and data were carefully analysed, the rational for the hypotheses and implications of this study need more attention.

Response: Thank you for reviewing our manuscript, as well as providing the constructive comments. We carefully revised the manuscript based on your comments. The rationale for the hypotheses and implications of this study have also been substantially revised. We hope the updated manuscript is satisfactory for publication.

Major concerns

(1) Firstly, as mentioned in the limitation of the study. this study is cross-sectional; the outcome measures might have affected the ratings of personality, moral, and ideology. It is recommended to include the follow-up data in addition to the current data set. 

Response: Thank you for the suggestion. Indeed, we are presently running multi-wave surveys, after the first data sampling, which was introduced in this manuscript. Considering that the pandemic in Japan is still active, we would like to publish research based on the longitudinal data as a different study. The purpose of this study is to provide immediate data collected at the beginning of the emergency situation. 

(2) Secondly, I am not sure if this study “can provide informative implications to the governance and management of COVID-19”. For example, “Individuals with increased tendency for neuroticism should be given extra care…” How can the government assess residents neuroticism, and how would they approach those people high in neuroticism? Similarly, I wonder how the government can reinforce conscientiousness and agreeableness, given that personality is a trait that wouldn’t change in the short period of time.

Response: Thank you for pointing this practical issue. We apologize for the misunderstanding regarding our implications in helping governance and management. Defining or changing the personality of each individual was not expected, because it is neither practical nor prior in the emergency situation of COVID-19, and as you cited, it is impossible to change people’s personality in a short time. We expected that our results could provide empirical evidence to the governance, and help them to implement certain general measures to reduce people’s tendency of neuroticism and improve their conscientiousness and agreeableness; for example, by encouraging people to take their own responsibility in preventing infection and epidemic in their daily life and to better understand and express gratitude to the efforts of other people. In addition, “Reinforce” was not the appropriate word. We revised this part with additional explanation in Lines 413–421 and Lines 458–463.

(3) Finally, I wonder now big five was selected as “personality” among other models. Moreover, it is not clear, in Introduction, how each personality may affect concern and consideration, preventive behavior, and knowledge acquisition of COVID-19. Please expand on the rational for the relationship between big 5 and those outcomes. On the other hand, the relationship between personality and mental health can be too obvious to replicate in this study (e.g. Table 2). 

Response: Thank you for citing these crucial questions. We selected the Big Five model among other models as we consider that it accurately represents the personality traits demonstrated by large cross-sectional samples and is widely accepted by not only academic but also industry and general public. In addition, the Big Five model has been well investigated and demonstrated in Japan, the country where we conducted our survey. We added explanation and related previous research in Lines 68–79. The issues in our hypotheses were also indicated by the other reviewer. We cannot rebuild our hypotheses after the results have been revealed (HARKing), including supposing how each personality may affect people’s concern and behavior ex post facto. Thus, we would like to represent our study as an explanatory research. The relevant revision can be found in Lines 101–114. 

Minor concerns

(1) I wonder how authors have developed the rational for hypothesis 2. To me the relationship would be the other way around. Related to the point above, mental health plays little role in this study if it is used as an outcome variable.

Response: Thank you for your suggestion. We reviewed our hypothesis 2, and as suggested by you, found that the relationship between people’s concerns and behavior with their mental health status was ambiguous and debatable. We excluded the contents related to this hypothesis and its results in the revised manuscript. 

(2) How was the target number (2000) agreed? Was it determined by the design of statistical analyses?

Response: We did a priori power analyses before the survey to decide the target number. In the revised version, we added the process of determining the target sample size, including the details of a priori power analyses by using G * Power (Lines 141–150).

***

Response to Reviewer 2

Opening comments

The author vigorously investigated the characteristics of citizens regarding COVID-19 factors. Outcomes will be beneficial for the future as intended by the author if appropriately performed. However, this study includes several issues to be discussed before verifying the results.

Response: Thank you for reviewing our manuscript, as well as pointing out the issues by sharing your informative and pertinent suggestions. Following your comments and suggestions, we carefully and substantially revised the manuscript. We hope the revised manuscript addresses your concerns and satisfies the requirements of publication. The manuscript has also been carefully reviewed by a native English speaker.

Detailed comments

(1) First, it was 7th April 2020, not March, that the Japanese government first declared a state of emergency against the 2020 COVID-190 pandemic. Was this study conducted on 8 – 9th March? The relationship between the date of survey and social events just before it is crucial to interpret the results. The author should look through the whole manuscript if there are any inconsistencies about chronological order.

Response: We are extremely sorry for this careless mistake. We conducted the survey on 8–9th of April, not March. We have revised all the inconsistencies of the date and carefully checked the descriptions in the revised manuscript.

(2) The author presented three hypotheses in the last paragraph of the introductory section. However, these ideas seem too vague to be defined as each hypothesis. Most results extracted from the obtained data are possibly consistent with these ideas. For example, H1: “Personality, morality, and political ideology can predict the mental health status, concern and consideration, preventive behavior, and knowledge acquisition of COVID-19” will be supported by the data suggesting any relationship between perceptual and behavioral characteristics. Therefore, I can hardly identify this study as a confirmatory one even if the author had developed hypotheses in advance. If the author had expected any specific results before conducting this survey, describe it, like that: “participants with liberal ideology should be more concerned about COVID-19.” Otherwise, this study should be presented as an exploratory study.

Response: Thank you for indicating this critical issue. We agree with your opinions, that the hypotheses were too vague to be defined. However, to assure that our study is not HARKing, we cannot change our hypotheses ex post facto, after revealing all the results. Thus, we decided to delete all hypotheses in the revised manuscript and change our study to an exploratory one. Please find the related revision in Lines 101–114, and Lines 381–384.

(3) Maybe relevant to the vagueness of the hypotheses, this manuscript is too long to be read straightforward. Readers hardly understand what is the point of the study. I recommend the author to examine the whole manuscript to rewrite so that readers can focus on the critical findings.

Response: In the updated version, we revised the manuscript substantially and excluded the contents, which were not regarded as crucial findings. The major revisions were listed as below:

1. We deleted the part of COVID-19 knowledge test. The reason of deleting this part will be explained later, as response to the other comment from you (Detailed comments 5).

2. We deleted the contents related to Hypothesis 2 in the last version, about how the concern and behavior of COVID-19 patients affects their mental health status. This revision was based on the comments of the other reviewer.

3. We substantially reorganized and revised our results as well as the tables. For the results, besides the opening section of Data collection and demographics and Validation of scale reliability, the sections (subtitles) were reduced from 4 to 2. The tables in total were reduced from 11 to 5.

4. The contents of the third section of Results entitled “Influence of personality, morality, and ideology on mentality, opinion and behavior of the general public toward COVID-19,” which was our main result based on the regression analyses was generally reorganized (Lines 284–326).

(4) The author confirmed to have gain an informed consent from each participant in the first paragraph in the methods section. Did the participants send signed informed consent form? If so, the author should describe it as well as the way of storing the signed informed consent forms securely (actually, I guess not).

If this questionnaire survey was conducted with gathering the data anonymously, the researchers should have disclosed the study protocol and data using policy to the respondents before starting the questionnaire. In this case, respondents are deemed to have accepted the policy because they voluntarily sent their answers. It is not the same as written informed consent, but an acceptable way of study. If this study was conducted with this way, the author should describe it. Otherwise, gained data without any explanation to the respondents cannot be a material of scientific study of human subjects.

Response: We used the second method. We added description of the method as follows in Lines 147–150.

“The survey was conducted anonymously. The study protocol and data using policy were disclosed at the recruitment page as well as the beginning of the questionnaire. The questionnaire survey commenced only after the participant accepted the data using policy and agreed to participate.”

(5) The author examined the participant’s knowledge about COVID-19 with Q5 of the questionnaire. The structure of this section contains several issues.

- What was the purpose of these quiz?

- Considering that this series was firstly developed by the author, I guess the data has not been standardized. How the author qualified the score of this section?

- Some questions of the series seem hard to determine the correct answer. Did some authorities check which answer was correct?

- The author identified the answers of “do not know” in Q5 (knowledge test) as knowledge uncertainty. In my sense, however, there were few persons to choose “do not know” when they were actually uncertain about each question, guessing considerable amount of people choose “yes” or “no” with their instinct. Therefore, I do not believe that this variable represents knowledge uncertainty of the participants. Are there any preceding studies to support the way of the author?

Overall, I cannot justify the scientific use of the results of this section, unless these issues are entirely addressed.

Response: Thank you for the valuable suggestions. As aforementioned, we decided to delete this part in the revised version. This knowledge test on COVID-19 was originally created by one of the authors. He is a biologist, and consequently should be a person of authority to check the answers. However, besides the authority check, there were more issues, as you mentioned, when regarding it as a part of psychological questionnaires, including its standardization, response method, data analyses, as well as the scientific use of the results. We apologize that these issues were not handled before we decided to put this quiz to our questionnaire and conducted the survey.

(6) The author described that there were 264 unvaried responses. I was surprised to know the outcome that more than 10% of the data should be excluded. Although I have no evidence regarding the standard rate, 10% looks extraordinary (In my experience, approximately 2 – 3 %). Do the author this result can be acceptable? I am afraid the population of this survey (from Yahoo group) itself was contaminated, in other words including a considerable number of dishonest subjects. Can the author believe all adopted respondents answered the questionnaire sincerely?

Were there any solutions to exclude multiple posting of a certain person certainly? If not, the validity of this survey should be collapsed.

Response: For the method of data exclusion, we added details in Lines 221–224. To assure the data quality, we used serious criteria to exclude data. Normally, crowdsourcing surveys only exclude data based on the validation of attention check question (ACQ), but in our study, we used two more methods to detect the invalid data, by checking the consistency between residence place and postcode, and checking the SD of the answers in the same questionnaire page. SD check can detect the hidden dishonest or satisficing participants but causes larger number of data exclusion. In our study, the data from 264 participants were excluded due to the SD check. They gave the same response to all questions at least in one questionnaire page (in one questionnaire page, there were 15–20 questions). This was a serious criterion, because in total 10 questionnaire pages were set as the target of SD check (mentioned in the newly added footnote 2). In particular, from our results, we can find that the traditional ACQ (1.2% was excluded by ACQ) and simple validations such as postcode check (3.9% was excluded by postcode check) can be easily evaded by experienced satisficers (Crowdsourcing survey is not a new method, so there should be many experienced satisficers who can cleverly evade the ACQ and cheat in the survey. This issue has been discussed in the previous research we cited in Lines 222–223). To exclude the satisficing data and ensure the data quality as much as possible, was the main purpose of using such serious criteria in our study. We believe our dataset filtered by the serious exclusion was reliable, based on the following evidence:

1. We conducted validation of scale reliability. All scales of BFS, DASS, and MFQ were validated by using our dataset. If the dataset was inaccurate, the confirmatory factor analyses on these scales should be collapsed too.

2. The results of statistical analyses (regressions, t-test and ANOVA) were all explainable. If the dataset was contaminated, we should get more incomprehensible results.

For the possibility of multiple posting, Yahoo! Crowdsourcing has a filtering system to ensure that each user can only participate in the survey once. Because the survey was programmed by jsPsych and performed in our server (we added this detailed information in Lines 133–135), it was possible to define multiple posting by collecting participants’ IP addresses, but we did not do this due to privacy protection. 

(7) In summary, the major weaknesses of this manuscript are below:

- Doubtful subjects

- Unconcreted hypotheses

- Redundant presentation

Response: Thank you very much for summarizing these weak points in the last version of our manuscript. As stated above, we made substantial revision in our manuscript and added necessary explanations. We hope the revised version is clearer to express the significance of this study, easier to read and satisfactory for publication.

---

## [Decision Letter · Decision Letter 1]

15 Jun 2020

PONE-D-20-12844R1

Mentality and behavior in COVID-19 emergency status in Japan: Influence of personality, morality and ideology

PLOS ONE

Dear Dr. Qian,

Thank you for submitting your manuscript to PLOS ONE. After careful consideration, we feel that it has merit but does not fully meet PLOS ONE’s publication criteria as it currently stands. Therefore, we invite you to submit a revised version of the manuscript that addresses the points raised during the review process.

The reviewer addressed some concerns again. Please revise your manuscript carefully.

We look forward to receiving your revised manuscript.

Kind regards,

Kenji Hashimoto, PhD

Academic Editor

PLOS ONE

Reviewers' comments:

Reviewer's Responses to Questions

**Comments to the Author**

1. If the authors have adequately addressed your comments raised in a previous round of review and you feel that this manuscript is now acceptable for publication, you may indicate that here to bypass the “Comments to the Author” section, enter your conflict of interest statement in the “Confidential to Editor” section, and submit your "Accept" recommendation.

Reviewer #1: All comments have been addressed

Reviewer #2: All comments have been addressed

2. Is the manuscript technically sound, and do the data support the conclusions?

Reviewer #1: Yes

Reviewer #2: (No Response)

3. Has the statistical analysis been performed appropriately and rigorously? 

Reviewer #1: Yes

Reviewer #2: (No Response)

4. Have the authors made all data underlying the findings in their manuscript fully available?

Reviewer #1: Yes

Reviewer #2: (No Response)

5. Is the manuscript presented in an intelligible fashion and written in standard English?

Reviewer #1: Yes

Reviewer #2: (No Response)

6. Review Comments to the Author

Reviewer #1: Thanks for addressing all my comments. However, some responses were not yet sufficient to convince the reviewer that this manuscript is ready for publication.

I am not yet convinced by the rational for this study. The response says “our results could provide empirical evidence to the governance, and help them to implement certain general measures to reduce people’s tendency of neuroticism and improve their conscientiousness and agreeableness”

This still sounds that personalities can be changed by the government.

Moreover, it is claimed that “by encouraging people to take their own responsibility in preventing infection and epidemic in their daily life and to better understand and express gratitude to the efforts of other people”.

I do not think that government can change people’s personalities by those exercises. In addition, the government is still encouraging people to exercise these precautions, REGARDLESS their personalities.

Thus, it is not clear to me how the results of these studies can help the government. If the aim of this study is just to explore the relationships between variables, that would significantly reduce the value of the study. As I suggested, it would be better to at least include the follow up data should the results of this study cannot help the government and society.

Reviewer #2: (No Response)

7. PLOS authors have the option to publish the peer review history of their article (what does this mean?). If published, this will include your full peer review and any attached files.

Reviewer #1: No

Reviewer #2: Yes: Akihiro Shiina

---

## [Author Response · Author response to Decision Letter 1]

20 Jun 2020

Dear Reviewer,

Thank you very much for reviewing our manuscript. We have carefully revised our manuscript based on your constructive comments. We highlighted the changes made to the previous version in red in the file labeled 'Revised Manuscript with Track Changes'. Our point-by-point response is shown as below. We recommend checking our detailed response in the separated file labeled 'Response to Reviewers', which should be easier to read. The line numbers are based on the 'Revised Manuscript with Track Changes'.

We hope the revised manuscript is satisfactory for the publication in PLOS ONE.

Sincerely yours, 

Kun Qian, PhD

***

Comments: Thanks for addressing all my comments. However, some responses were not yet sufficient to convince the reviewer that this manuscript is ready for publication.

I am not yet convinced by the rational for this study. The response says “our results could provide empirical evidence to the governance, and help them to implement certain general measures to reduce people’s tendency of neuroticism and improve their conscientiousness and agreeableness”

This still sounds that personalities can be changed by the government.

Moreover, it is claimed that “by encouraging people to take their own responsibility in preventing infection and epidemic in their daily life and to better understand and express gratitude to the efforts of other people”.

I do not think that government can change people’s personalities by those exercises. In addition, the government is still encouraging people to exercise these precautions, REGARDLESS their personalities.

Thus, it is not clear to me how the results of these studies can help the government. If the aim of this study is just to explore the relationships between variables, that would significantly reduce the value of the study. As I suggested, it would be better to at least include the follow up data should the results of this study cannot help the government and society.

Response: Thank you for your comments and concerns. To be very clear, we do not consider that the government should change people’s personalities. Considering your comments, we revised the relevant descriptions by emphasizing the minds and behaviors associated with neuroticism, conscientiousness, and agreeableness (Line 407-414). We also revised the passage regarding governance and explained that the government should focus on personality traits as well as personality-related thinking and behaviors, rather than focusing on specific individuals or groups of individuals who display prominent personality traits of concern (Line 414-423). Furthermore, please consider that the implications to governance were not only about personality (Line 407-423) but also concerned morality, ideology, and comparisons based on social demographic data (Line 424-445). The implications from the viewpoint of personality was just one aspect of this paragraph. 

We hope we have made the above points clear with the minor revision. With respect to the follow-up data, as mentioned in our last reply, we are now collecting longitudinal data through weekly wave surveys. The COVID-19 pandemic remains active. Thus, we would like to publish the longitudinal data as a different paper. The purpose of this study was to provide immediate data collected at the beginning of the outbreak.

---

## [Decision Letter · Decision Letter 2]

25 Jun 2020

Mentality and behavior in COVID-19 emergency status in Japan: Influence of personality, morality and ideology

PONE-D-20-12844R2

Dear Dr. Qian,

We’re pleased to inform you that your manuscript has been judged scientifically suitable for publication and will be formally accepted for publication once it meets all outstanding technical requirements.

Kind regards,

Kenji Hashimoto, PhD

Section Editor

PLOS ONE

Additional Editor Comments (optional):

Reviewers' comments:

Reviewer's Responses to Questions

**Comments to the Author**

1. If the authors have adequately addressed your comments raised in a previous round of review and you feel that this manuscript is now acceptable for publication, you may indicate that here to bypass the “Comments to the Author” section, enter your conflict of interest statement in the “Confidential to Editor” section, and submit your "Accept" recommendation.

Reviewer #1: All comments have been addressed

2. Is the manuscript technically sound, and do the data support the conclusions?

Reviewer #1: Yes

3. Has the statistical analysis been performed appropriately and rigorously? 

Reviewer #1: Yes

4. Have the authors made all data underlying the findings in their manuscript fully available?

Reviewer #1: Yes

5. Is the manuscript presented in an intelligible fashion and written in standard English?

Reviewer #1: Yes

6. Review Comments to the Author

Reviewer #1: (No Response)

7. PLOS authors have the option to publish the peer review history of their article (what does this mean?). If published, this will include your full peer review and any attached files.

Reviewer #1: No

---

## [Editor Report · Acceptance letter]

30 Jun 2020

PONE-D-20-12844R2 

Mentality and behavior in COVID-19 emergency status in Japan: Influence of personality, morality and ideology 

Dear Dr. Qian:

I'm pleased to inform you that your manuscript has been deemed suitable for publication in PLOS ONE. Congratulations! Your manuscript is now with our production department. 

Kind regards, 

on behalf of

Prof. Kenji Hashimoto 

Section Editor

PLOS ONE